# Unsteady Flow Process in Mixed Waterjet Propulsion Pumps with Nozzle Based on Computational Fluid Dynamics

**Can Luo [1,2,*], Hao Liu [1], Li Cheng [1,*], Chuan Wang [1], Weixuan Jiao [1] and Di Zhang [1]**

[1] College of Hydraulic Science and Engineering, Yangzhou University, Yangzhou 225009, China; yzuliuhao@163.com (H.L.); wangchuan198710@126.com (C.W.); DX120170049@yzu.edu.cn (W.J.); DX120180054@yzu.edu.cn (D.Z.)

[2] Key Laboratory of Fluid and Power Machinery of Ministry of Education, Xihua University, Chengdu 610039, China

* Correspondence: luocan@yzu.edu.cn (C.L.); chengli@yzu.edu.cn (L.C.)

**Abstract:** The unsteady flow process of waterjet pumps is related to the comprehensive performance and phenomenon of rotating stall and cavitation. To analyze the unsteady flow process on the unsteady condition, a computational domain containing nozzle, impeller, outlet guide vane (OGV), and shaft is established. The surface vortex of the blade is unstable at the valley point of the hydraulic unstable zone. The vortex core and morphological characteristics of the vortex will change in a small range with time. The flow of the best efficiency point and the start point of the hydraulic unstable zone on each turbo surface is relatively stable. At the valley point of the hydraulic unstable zone, the flow and pressure fields are unstable, which causes the flow on each turbo surface to change with time. The hydraulic performance parameters are measured by establishing the double cycle test loop of a waterjet propulsion device compared with numerical simulated data. The verification results show that the numerical simulation method is credible. In this paper, the outcome is helpful to comprehend the unsteady flow mechanism in the pump of waterjet propulsion devices, and improve and benefit their design and comprehensive performance.

**Keywords:** waterjet propulsion pump; unsteady flow process; test; computational fluid dynamics

## 1. Introduction

Unlike a propeller device, the waterjet propulsion device enables the ship to obtain navigational power by utilizing the reaction force of the water flow ejected by the propulsion pump. The waterjet propulsion device has the advantages of flexible operation, excellent maneuverability, high speed, outstanding anti-cavitation performance, and high efficiency [1]. The propulsion pump is well-protected for being arranged inside. When the ship speed exceeds 25 knots, the total efficiency of the waterjet propulsion device can reach more than 60%. Based on these advantages, the waterjet propulsion device has been widely used in high-speed performance vessels [2,3]. As the core component of the waterjet propulsion device, the performance of the propulsion pump is related to the performance of the entire device, and the nozzle is also an important part. Considering the thrust and layout requirements, the guide vane mixed-flow pump and axial flow pump are generally adopted in current waterjet propulsion vessels. In addition, other types of pump such as screw pump and permanent maglev (shaft less) pump utilized in the waterjet propulsion device are in the experimental research stage [4,5]. The pump employed in the waterjet propulsion device of this research is the guide vane mixed-flow pump. Moreover, pumps are essential for all animals. The heart tube of the embryonic vertebrate has been described as a peristaltic pump before the development of discernable chambers and valves at

these early stages [6]. Then the bioinspired valveless pump is designed related to the pumping process of the heart tube and applied in the fields, such as microfluidics, drug delivery, biomedical devices, and cardiovascular pumping systems, becoming an important topic nowadays [7].

In recent years, simulation methods, such as CFD (Computational Fluid Dynamics) and control system simulation, combined with experiment technology, has been applied in numerous fields such as noise [8], vibration [9–13], turbo machinery [14–20], valve [21,22], jet flow [23,24], heat transfer [25], and hydraulic systems [26]. The researchers not only carried out research on waterjet propulsion devices from both theoretical and experimental aspects, but also conducted research works on waterjet propulsion devices by means of CFD technology. By using CFD technology, Ahn predicted the performance of the designed mixed-flow pump. The results are well agreed with the measured data in the cavitation tunnel test [27]. Wu et al. compared and analyzed the cavitation in the tip clearance of propulsion pump at different revolutions utilizing LDV (Laser Doppler Velocimetry) test technology and high-speed photography technology [28]. Tang analyzed the influence of the guide vane on the performance of the axial flow pump, adopting a numerical simulation and model test. A fixed axial clearance value between the impeller and the vane was concluded. When the value was exceeded, the influence on the performance of impeller was negligible [29]. Duerr analyzed the characteristics of the waterjet propulsion device, applying the non-uniform inflow condition [30]. Verbeek and Bulten focused on the uniformity of the flow field in front of the impeller, and analyzed the effects of boundary laminar flow and turbulence intensity on the uniformity of the flow field [31,32]. Brandner and Walker used pressure probes and visual test methods to conduct quantitative and qualitative experimental studies on the waterjet propulsion flush inlet. It was found that cavitation occurred in the lips in a wide range of operating conditions [33,34]. Park conducted a model test on the influent runner model at the wind tunnel laboratory [35,36]. Gong et al. simulated the flow in the entire waterjet propulsion device with unsteady methods and obtained the changes of the free surface in the waterjet propulsion device at different times [37]. Cao et al. compared the effects of uniform inflow and non-uniform inflow on the performance of the waterjet propulsion pump [38]. Zhang et al. analyzed the effect of cavitation on the thrust performance of the nozzle with CFD technology [39]. Cheng and Xia et al. studied the rotation stall existing in the propulsion pump and proposed corresponding suppression measures [40,41]. Recently, works have mainly focused on the property of the waterjet propulsion device, but research on the unsteady flow process between the nozzle and the propulsion pump has not been seen yet. In this study, the hydraulic performance and unsteady flow process of the propulsion pump with nozzle will be analyzed by using CFD technology, and the numerical simulation results are verified compared with the experiment result on the model test.

## 2. Numerical Simulation Method

### 2.1. Geometry Model

A computational domain geometric model including the nozzle is established to obtain stable flow field data of the mixed-flow waterjet propulsion device. Along the flow direction, subdomains are the prolonged inlet section, the inlet transition section, the impeller, the guide vane (GV), the nozzle, and the prolonged outlet section, as shown in Figure 1.

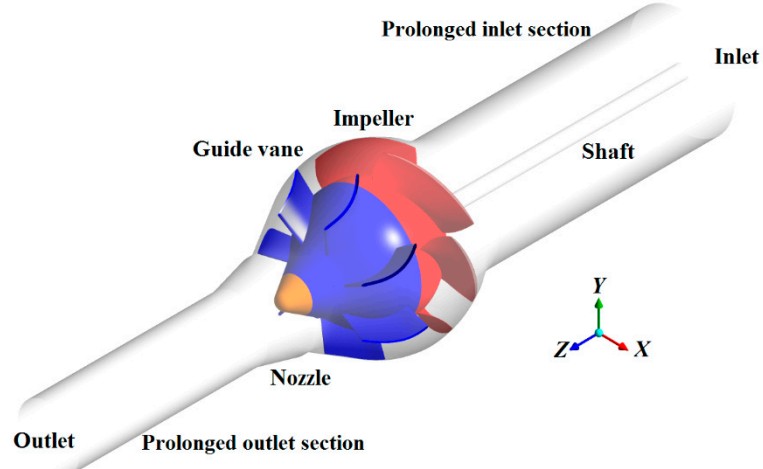

**Figure 1.** Computational domain of mixed waterjet pump.

## 2.2. Mesh Generation and Independence Analysis

According to the geometrical characteristics of each sub-domain, different meshing strategies are employed to construct the discrete meshes of each sub-domain. The impeller, the GV and the nozzle adopt the J-type, H-type, and O-type meshing methods, separately. By using ANSYS-ICEM, a reasonable block structure for each sub-domain is created to generate the number of structured meshes by controlling the number of mesh nodes and the growth rate. The accuracy of the calculation results will be poor if the meshes do not meet the calculation requirements. However, if the number of meshes is too large, it will occupy numerous computing resources. Therefore, the mesh independence analysis needs to be performed. The number of impeller meshes is 0.31 million, 0.5 million, 0.7 million, 0.89 million, and 1.1 million, recorded as mesh 1–5, as shown in Figure 2.

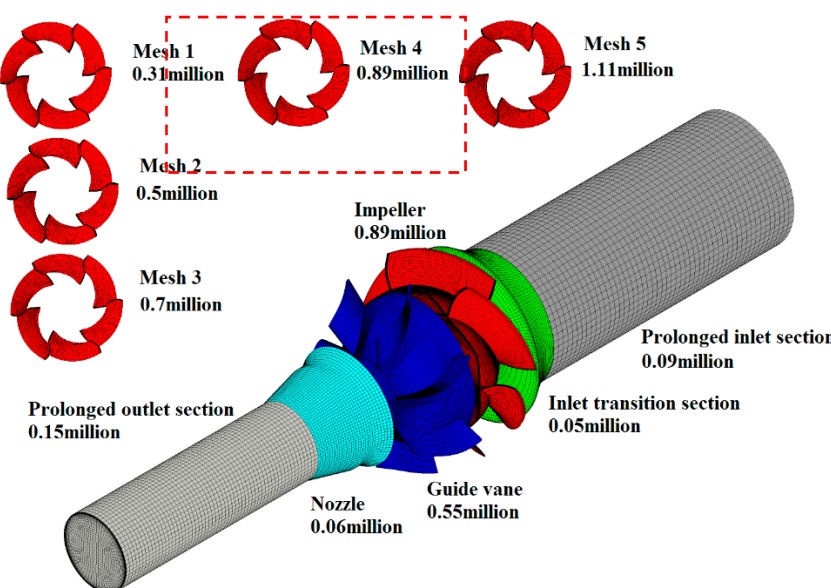

**Figure 2.** Meshes of computational domain.

The head $H$ and efficiency $\eta$ is calculated by Equations (1) and (2) when the flow rate is $Q_{\text{BEP}}$, 1.13 $Q_{\text{BEP}}$ and 1.33 $Q_{\text{BEP}}$. The relative head $H'$ and relative efficiency $\eta'$ are calculated by dividing the head and efficiency of mesh 4

$$H = (P_{\text{int}} - P_{outt})/\rho g \tag{1}$$

$$\eta = \frac{\rho g Q H}{P_{shaft}} \tag{2}$$

where $\rho$ is the density of water and the value is $10^3$ kg/m$^3$, $g$ is the acceleration due to gravity and the value is 9.81 m/s$^2$, $H$ is the total head of the propulsion pump in m, $P_{int}$ is the total pressure at the entrance of the propulsion pump in Pa, $P_{outt}$ is the total pressure on the outlet of the pump in Pa, $P_{shaft}$ is the shaft power in kW, and $\eta$ is the efficiency.

The relative head and relative efficiency are drawn in Figure 3. When the flow rate is $Q_{BEP}$, the relative head $H'$ of each mesh scheme has a certain difference, but the relative efficiency $\eta'$ is basically consistent. When the flow rate is 1.33 $Q_{BEP}$, the relative head and the relative efficiency $\eta'$ has a larger difference, especially mesh 1 and mesh 2. The disparity of mesh 3 tends to decrease. As the number of meshes increases, both the relative head $H'$ and the relative efficiency $\eta'$ gradually enhance, and this trend exceeds to become clearer as the flow rate increases. When the number of meshes surpass 1.79 million, the relative head $H'$ and the relative efficiency $\eta'$ tend to be constant. Therefore, mesh 4 is chosen as the final mesh, as shown in Figure 2.

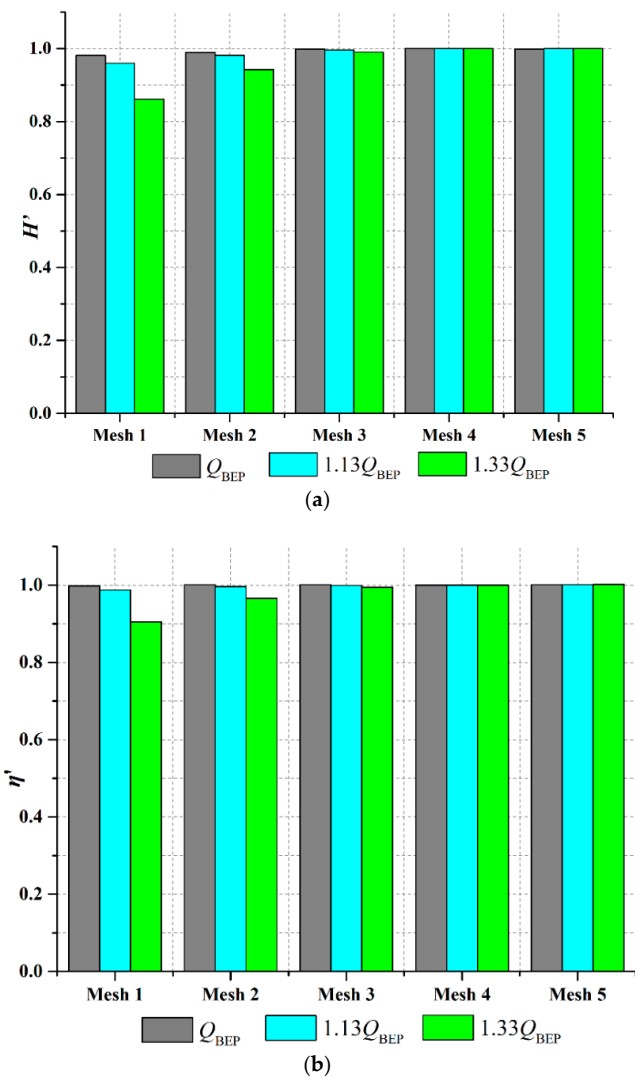

**Figure 3.** Mesh independence analysis. (**a**) Histogram of $H'$ for each mesh at $Q_{BEP}$, 1.13 $Q_{BEP}$, and 1.33 $Q_{BEP}$. The relative head $H'$ of mesh 4 tends to be constant. (**b**) Histogram of $\eta'$ for each mesh at $Q_{BEP}$, 1.13 $Q_{BEP}$, and 1.33 $Q_{BEP}$. The relative head $\eta'$ of mesh 4 tends to be constant.

### 2.3. Numerical Methodolgy and Boundary Conditions

As heat transfer does not exist in the flow process of waterjet propulsion device, Navier–Stokes equations are utilized as the governing equations to describe the flow, and the finite volume method is also applied to calculate the flow in the computational domains. Considering the incompressible flow, the inlet is set as mass flow. The impeller is the rotating domain and the spinning speed is 700 r/min or 73.27 rad/s. The reference pressure is 1 atm. The outlet is set as average static pressure. In order to guarantee the value transfer, the interfaces between each sub-domain are set, in which interfaces on the inlet and outlet of the impeller are transient rotor stator, and the remaining interfaces are general connection. Scalable wall function is processed on the wall. The standard $k$-$\varepsilon$ turbulence model and the first-order upwind scheme are adopted. The convergence accuracy is $10^{-5}$. According to the rotating speed of the waterjet propulsion pump, the time step is 0.00023381 s and the total time is 0.685728 s. The corresponding transient turbulence convergence sample is obtained when the impeller rotates per degree. The transient turbulent convergence sample set of all time steps is obtained when the set time finishes. Usually, the numerous samples will increase the costing time and the amount of data. Therefore, this paper sets 36 samples per period; that is, the data is stored when the impeller rotated per ten degrees. Finally, 288 sample results were saved in 8 periods.

### 2.4. Pressure Pulsation Monitoring Probe Arrangement

Pressure pulsation can be seen as the difference between the pressure amplitude at different points in time and the average pressure amplitude over the entire time period. Pressure pulsation can usually be classified from pulsation performance and frequency. According to the pulsation performance, pressure pulsation can be divided into turbulent pressure pulsation, which ignores fluid compressibility, and pulse source pressure pulsation, which ignores fluid viscosity. Based on the pulsation frequency, the pressure pulsation can be divided into irregular random pressure pulsation, axial frequency pulsation, and blade frequency pulsation—the latter two pulsations are referred to as regular pressure pulsation. For random pressure pulsation, there are various induced factors, such as cavitation, secondary flow, non-uniform inflow, etc. In terms of the regular pressure pulsation, the main induced factors are the impeller rotation, the pump shaft rotation, and rotor–stator interaction, which are related to the axial frequency and the blade frequency, respectively. Generally, the blade frequency-related pressure pulsation is captured in the impeller. If the pressure pulsation on the upstream and downstream near the impeller shows the discipline, the rotating impeller has an effect on the flow there. The observed pressure pulsation characteristics are distinct for different pumps and for the same pump. The observed pressure pulsation characteristics are diverse under various operating conditions and monitoring positions.

Fifteen monitoring probes in the waterjet propulsion pump are shown in Figure 4, and the locations of each probe are listed in Table 1.

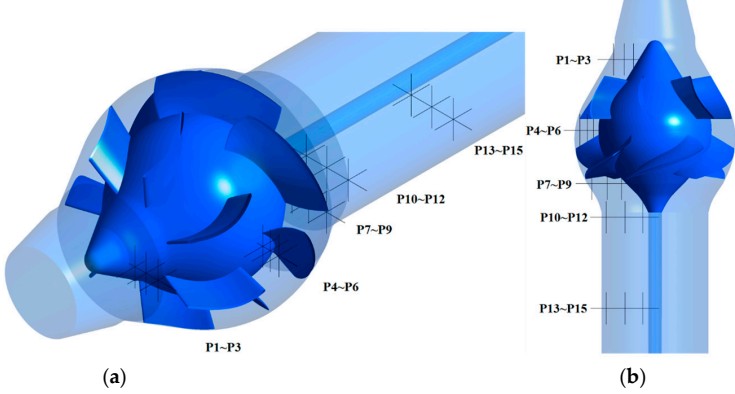

**(a)**        **(b)**

**Figure 4.** Diagram of pressure pulsation monitoring probe arrangement. (**a**) Lateral view. (**b**) Top view.

**Table 1.** Locations of each probe.

| Number | Location |
| --- | --- |
| P1–P3 | Outlet of guide vane (GV) |
| P4–P6 | Outlet of impeller |
| P7–P9 | Inlet of impeller |
| P10–P12 | Outlet of the prolonged inlet section |
| P13–P15 | In the prolonged inlet section |

Fifteen monitoring probes as above are pre-set in the pre-processing. Data of monitoring parameters are stored when the impeller rotates per degree. 2880 groups of data are obtained. While obvious regular periodic diversification trend starts from the second period in the unsteady calculation process, data of the last six periods, which contains 2160 groups of data, are chosen to execute FFT transformation. The data of the eighth period on each monitoring probes are plotted into the time domain chart of pressure pulsation.

Currently, the time domain analysis method and frequency domain analysis method are the main methods in studying the pressure pulsation. The time domain chart is utilized in the time domain analysis method. In the chart, the abscissa is the time-related parameters, such as the time and period, and the ordinate is the pressure. Frequency domain analysis method converts the irregular pressure pulsation into the superposition of a simple harmonic wave with different frequencies, amplitudes, and phases by performing FFT transformation. These two methods have their own advantages. The time domain chart can intuitively reflect the change of pressure pulsation on the monitoring probe with time. The frequency domain analysis illustrates the main pulsating component of the pressure pulsation and the primary factor affecting the pressure pulsation directly. The pressure pulsation discipline of each monitoring probe on condition A, B, and C will be analyzed by using the methods above.

For the impeller rotating at 700 rev/min, the shaft frequency is 11.67 Hz by using $f_z$ = n/60, the blade frequency is 70 Hz by using $f_b = 6f_z$, and multiples of shaft frequency are defined as $T_f$.

Pressure pulsation coefficient $C_p$ is introduced to analyze the pressure pulsation characteristic of each monitoring probes, and the pressure pulsation coefficient $C_p$ is calculated by applying Equation (3)

$$C_p = \frac{(p - \bar{p})}{0.5\rho V^2} \tag{3}$$

where $p$ is the instantaneous pressure in kPa, $\bar{p}$ is the time-averaged pressure in kPa, $\rho$ is the density of water in kg/m$^3$, and $V$ is the blade tip speed at the entrance of propulsion pump in m/s.

### 3. Test Arrangement and Verification

As shown in Figure 5, double cycle waterjet propulsion test bench is established to verify the reliability of the numerical simulation method. The test device consists of two cycles—the main cycle and the secondary cycle. The main cycle, which is applied to provide the navigation speed for the waterjet propulsion pump, consists of the centrifugal auxiliary pump, electromagnetic flow meter, butterfly valve, expansion joint, rectifying device, and piping system. The secondary cycle, which is used to test the hydraulic performance, includes the test zone (mixed pump), electromagnetic flow meter, butterfly valve, and piping system. The flow rate is measured by the flow meter located in the main cycle and secondary cycle. The head is obtained by calculating the pressure measured at the beginning of the inlet flow tube and the outlet pressure measuring tube. The shaft power is calculated from the test data of the torque meter. The comprehensive error of this test bench is ±1.33%.

Numerical simulation and model test are carried out when the rotational speed is 400 rev/min. Dimensionless head $H'$ and efficiency $\eta'$ are obtained by dividing the head and efficiency of the best

efficiency point, plotted in Figure 6. The CFD result shows a consistent trend with the test result. Therefore, the numerical method is reliable and suitable for the following research work.

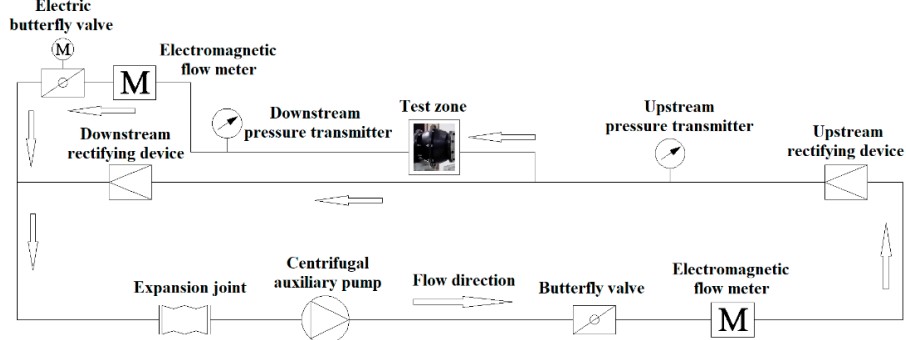

**Figure 5.** Double cycle waterjet propulsion test bench.

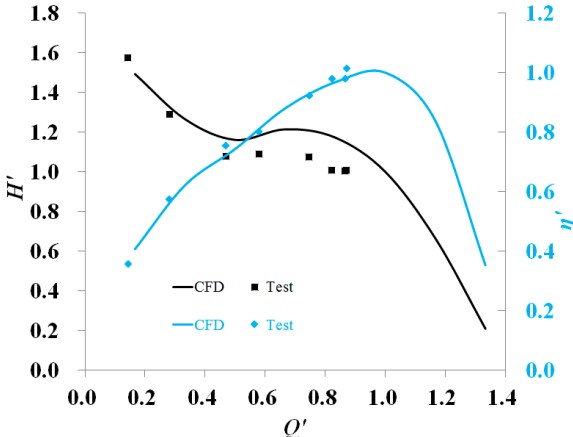

**Figure 6.** Contrast of CFD result and test result.

## 4. Results

### 4.1. Hydraulic Performance Curve

By diving the flow rate, head, and efficiency of the best efficiency point, dimensionless processing is performed for each condition. The dimensionless data is plotted into dimensionless a 'flow rate-head' curve and dimensionless 'flow rate-efficiency' curve, shown in Figure 7, in which the abscissa is $Q'$, the left ordinate, is $H'$ and the right ordinate is $\eta'$. The best efficiency point is marked as condition A. Under the condition B, the propulsion pump enters the hydraulic unstable zone. Condition C is the valley point of the hydraulic unstable zone.

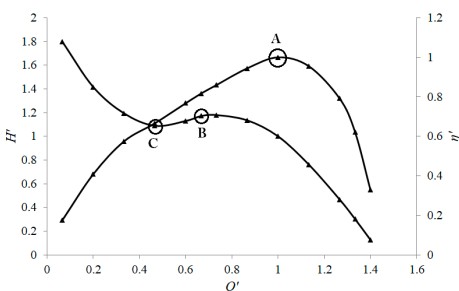

**Figure 7.** External characteristic curve.

## 4.2. Unsteady Flow Characteristics

Condition A, condition B, and condition C are chosen to be calculated unsteadily and analyzed. Figures 8–10 list the surface streamline on the blade of three time points (0T, 1/3T, and 2/3T) in the same period for condition A, B, and C. Under the condition A, surface streamlines on the suction side of each blade are smooth and the velocity at the leading edge is high. In the same period, surface streamline on the blade is stable with the lapse of time, which indicates that the pressure field and velocity field near the blade are stable under this condition. Under condition B, the stream at the leading edge flows to the trailing edge and the tip of the blade. The pressure field and velocity field near the blade are still stable. Under condition C, an evident vortex happens on the surface of the blade and the radial location of core vortex is span = 0.65. The size and shape of the vortex change with time, which indicates that the pressure field and velocity field near the blade are unstable.

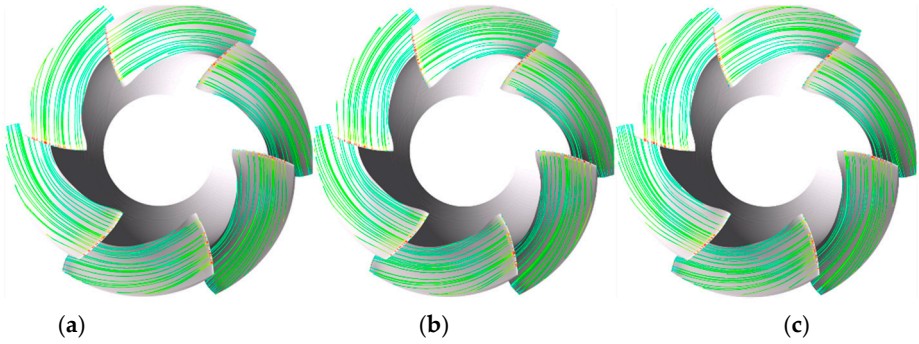

**Figure 8.** Streamline on the blade of different times for condition A (best efficiency point). (**a**) 0. (**b**) 1/3T. (**c**) 2/3T.

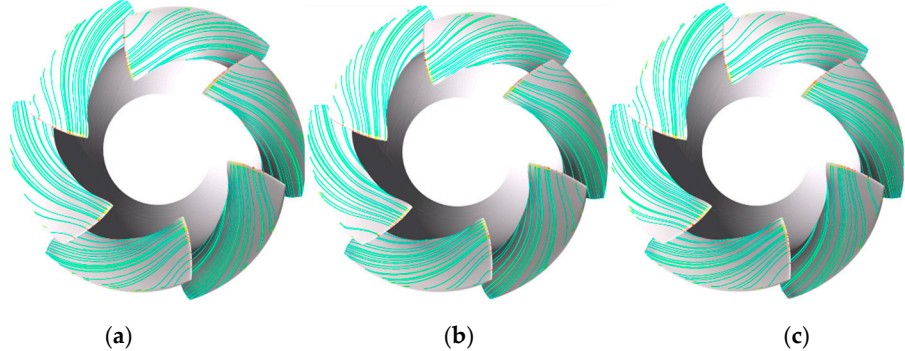

**Figure 9.** Streamline on the blade of different times for condition B (start point of hydraulic unstable zone). (**a**) 0. (**b**) 1/3T. (**c**) 2/3T.

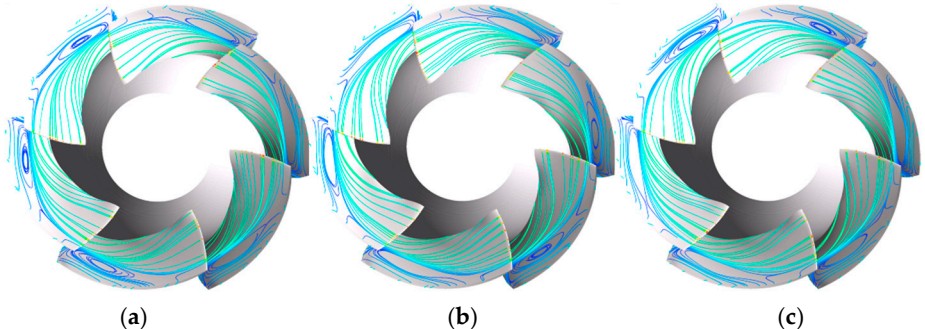

**Figure 10.** Streamline on the blade of different times for condition C (valley point of hydraulic unstable zone). (**a**) 0. (**b**) 1/3T. (**c**) 2/3T.

Three turbo surfaces from the hub to the shroud are sliced and recorded as TS1 (span = 0.1) near the hub, TS2 (span = 0.65) near the intermediate surface, and TS3 (span = 0.96) near the shroud, as shown in Figure 11.

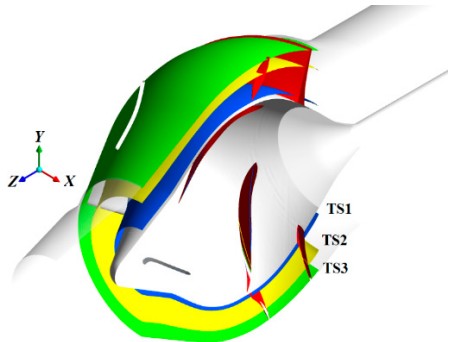

**Figure 11.** Turbo surfaces of the impeller.

Figures 12–14 are the streamlines on turbo surfaces TS1 under the condition A, B, and C, in which the velocity in the impeller is the relative velocity and the velocity in the GV is the absolute velocity. Under condition A, the streamlines are steady and vary slightly with time on the turbo surfaces of the impeller and GV. The velocity is rapid on the suction side of the impeller. The inlet attack angle is basically identical to the angle of the leading edge of the airfoil, which results in the excellent inflow condition and smooth stream in the blade-to-blade passage. A small-scale spanning vortex occurs at the tailing edge of the suction. Owing to the adjustment of the GV and the location of the vortex away from the impeller, the geometric shape and magnitude of the vortex shows no evident relationship with the time. Under condition B, slight deviation exists between the inlet attack angle and the airfoil angle of the blade in the impeller and the GV. The low-velocity region of the pressure surface of the leading edge of the impeller blade is enlarged. A large-scale spanning vortex occurs in each groove of GV, which extends from the inlet to the outlet in the axial direction, and occupies about 1/3 of the groove in the spanning direction. The streamlines of other parts in the groove are severely skewed and then gather near the outlet of the GV because of the spanning vortex. Under condition C, the smooth streamline in the groove is mildly affected. A distinct spanning vortex in the groove still exists and covers half of the groove on the spanning direction. The status and range of spanning vortex are basically maintained; however, the vortex core migrates in a small scale and the vortex status modifies, meaning the flow characteristic of GV is unstable.

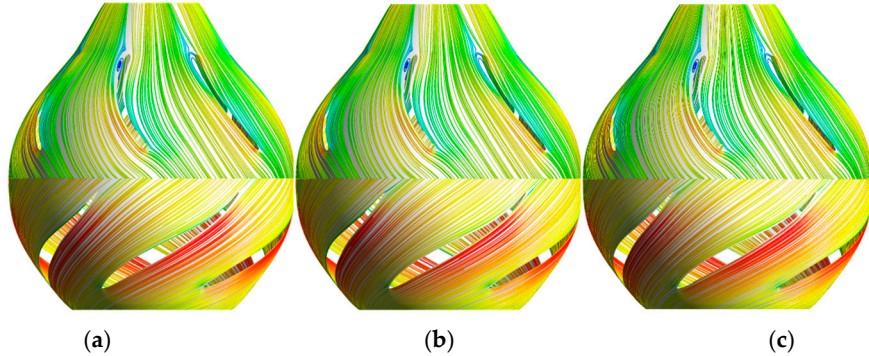

(**a**)　　　　　　　　　　　　　　(**b**)　　　　　　　　　　　　　　(**c**)

**Figure 12.** Streamlines on the turbo surface TS1 of condition A (best efficiency point). (**a**) 0. (**b**) 1/3T. (**c**) 2/3T.

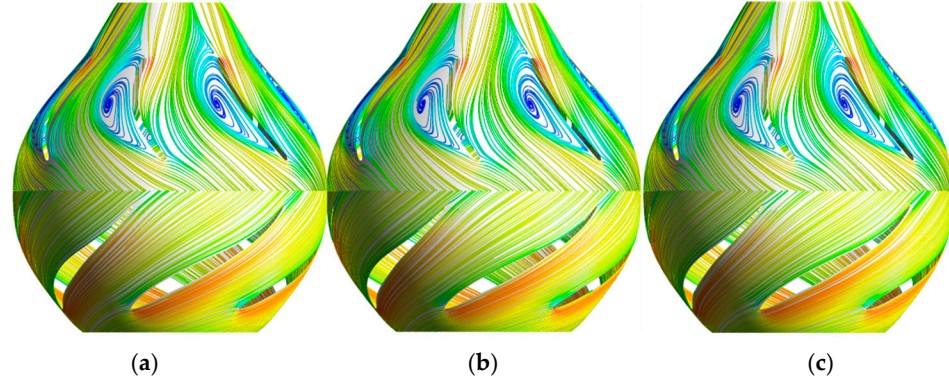

**Figure 13.** Streamlines on the turbo surface TS1 of condition B (start point of hydraulic unstable zone).
(**a**) 0. (**b**) 1/3T. (**c**) 2/3T.

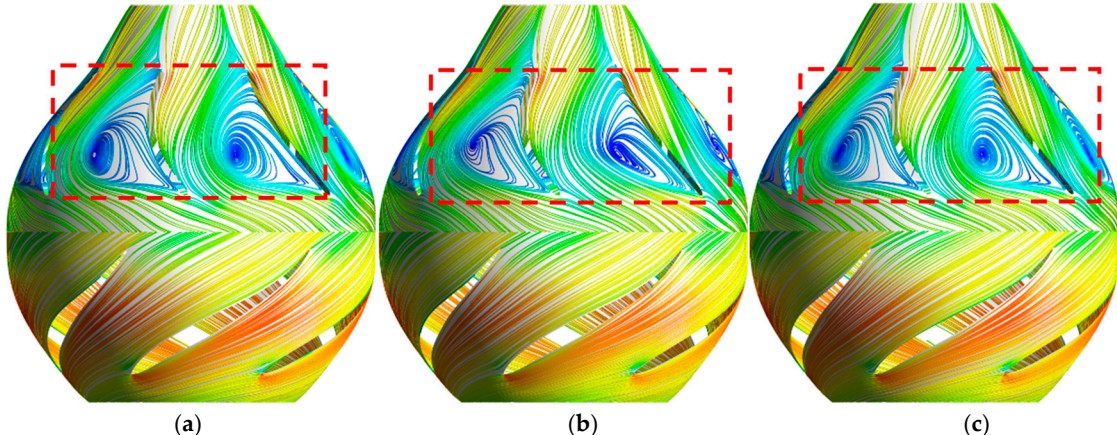

**Figure 14.** Streamlines on the turbo surface TS1 of condition C (valley point of hydraulic unstable zone). (**a**). 0 (**b**) 1/3T. (**c**) 2/3T.

Figures 15–17 are the streamlines on turbo surfaces TS2 under condition A, B, and C. Under condition A, the streamlines are smooth and no vortex occurs in the blade-to-blade passage of the impeller and GV. Under condition B, a slight deviation also occurs, but the streamlines are smooth in the groove of the impeller. As the arrows indicate in Figure 16, the streamlines deviate slightly near the tailing edge on the suction side of the impeller. The spanning vortex disappears in the groove of GV, but the shedding vortex is observed on the tailing edge of the outlet in the impeller. Part of the streamlines are severely skewed and have little impact on the mainstream. Under condition C, the inlet attack angle is consistent with the airfoil angle of the blade in the impeller. A distinct spanning vortex is observed near the tailing of different blades. The variation between the inlet attack angle of GV and the airfoil angle of the blade is apparent. Three vortexes marked as SV1, SV2, and SV3 occur on the spanning surface of GV. SV1 and SV2, located at the head edge and tailing edge of suction side, are in the channel of groove and rotate in clockwise. The range of SV1 is much larger than SV2. SV3 is the shedding vortex and located at the tailing of GV twirls on the opposite direction of the spanning vortex. As time passes, the shape and range of SV1, SV2, and SV3 varies.

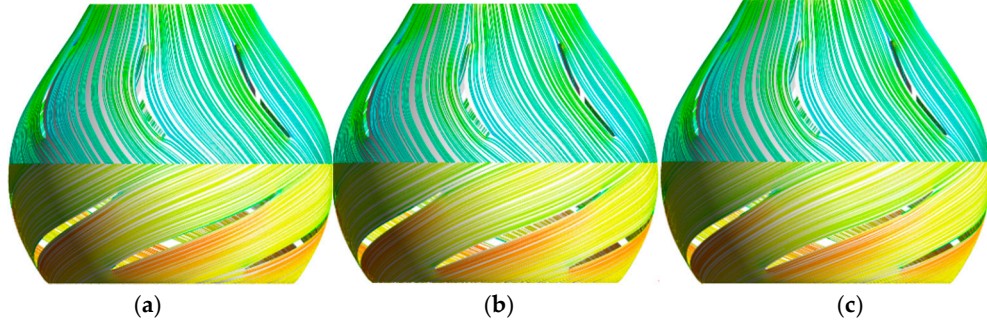

**Figure 15.** Streamlines on the turbo surface TS2 of condition A (best efficiency point). (**a**) 0. (**b**) 1/3T. (**c**) 2/3T.

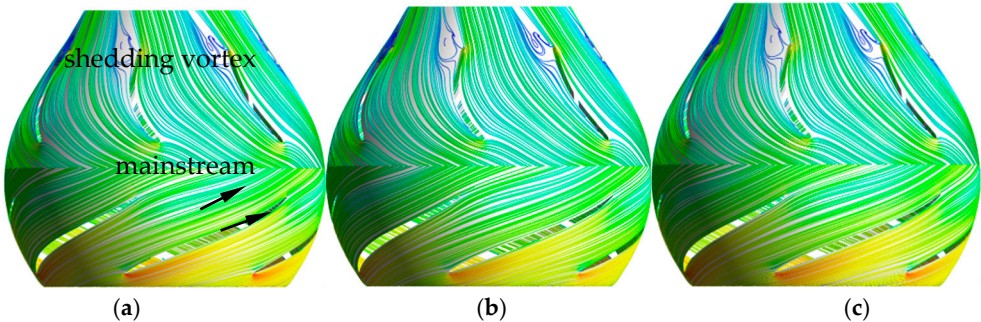

**Figure 16.** Streamlines on the turbo surface TS2 of condition B (start point of hydraulic unstable zone). (**a**) 0. (**b**) 1/3T. (**c**) 2/3T.

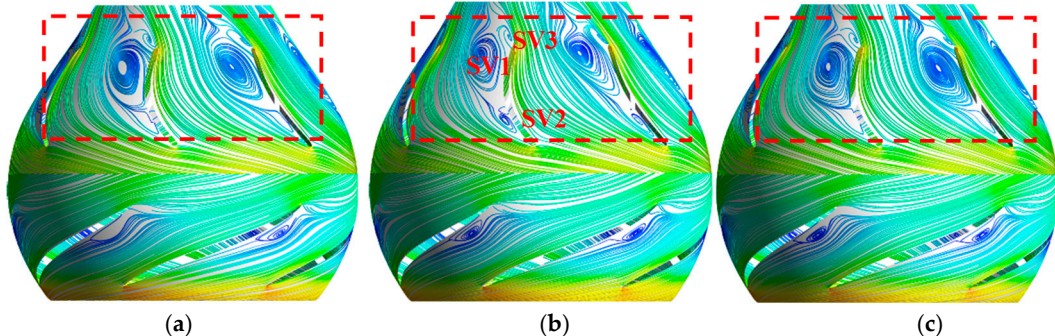

**Figure 17.** Streamlines on the turbo surface TS2 of condition C (valley point of hydraulic unstable zone). (**a**) 0. (**b**) 1/3T. (**c**) 2/3T.

Figures 18–20 are the streamlines on turbo surfaces TS2 under condition A, B, and C. Under condition A, the inlet attack angle is inconsistent with the angle of the leading edge of the airfoil. In one blade-to-blade passage, part of the stream flows from the suction side to the pressure side, then out of the blade-to-blade passage along the pressure side and the streamlines converge at the tailing edge of airfoil. The flow pattern in the blade-to-blade passage of GV is similar to the impeller. Under condition B, the stream at the entrance of the impeller flows into the neighboring groove on the opposite rotating direction instead of the GV. Part of the stream at the head edge on the suction side flows to the head edge on the pressure side of the neighboring blade and then out of the groove and into the neighboring groove after being dragged by the high-speed steam at the entrance of the groove. A low-velocity zone exists in the groove of the impeller, and the streamline is disordered. Under condition C, the distinction is huge between the inlet attack angle of the impeller and the airfoil angle of the blade, and an obvious spanning vortex appears at the head edge of the pressure side and the tailing edge of the suction side. Both spanning vortexes nearly cover the whole channel of the

groove in the spanning direction. The shape of spanning vortex will change with time, but the vortex core will not migrate and the location maintains.

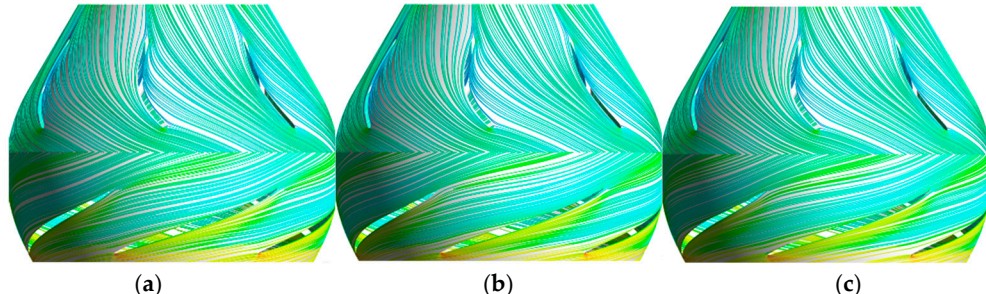

(**a**) (**b**) (**c**)

**Figure 18.** Streamlines on the turbo surface TS3 of condition A (best efficiency point). (**a**) 0. (**b**) 1/3T. (**c**) 2/3T.

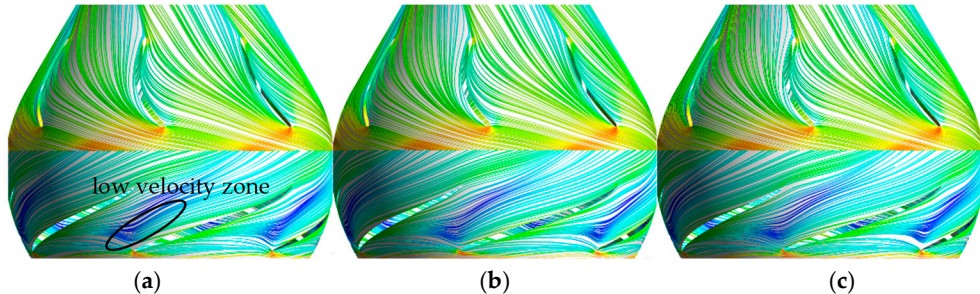

(**a**) (**b**) (**c**)

**Figure 19.** Streamlines on the turbo surface TS3 of condition B (start point of hydraulic unstable zone). (**a**) 0. (**b**) 1/3T. (**c**) 2/3T.

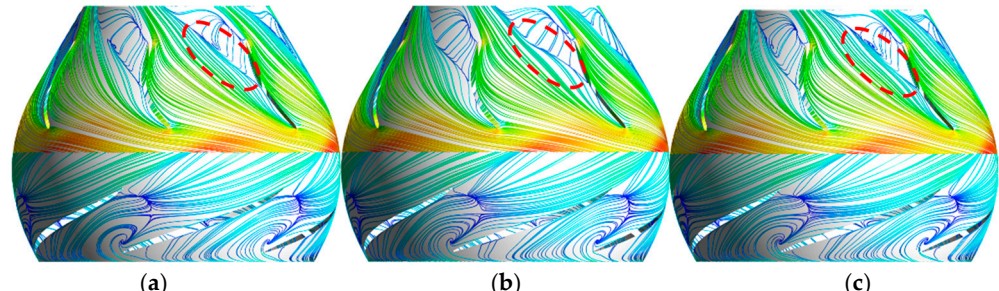

(**a**) (**b**) (**c**)

**Figure 20.** Streamlines on the turbo surface TS3 of condition C (valley point of hydraulic unstable zone). (**a**) 0. (**b**) 1/3T. (**c**) 2/3T.

*4.3. Pressure Pulsation*

Figure 21 shows the pressure pulsation time domain diagram and comparison of pressure pulsation amplitude of the monitoring probes P13–P15 in a period under condition A, B, and C. The main frequency, the secondary frequency, and the corresponding pressure amplitude of the monitoring probes P13–P15 under condition A, B, and C are listed in Table 2.

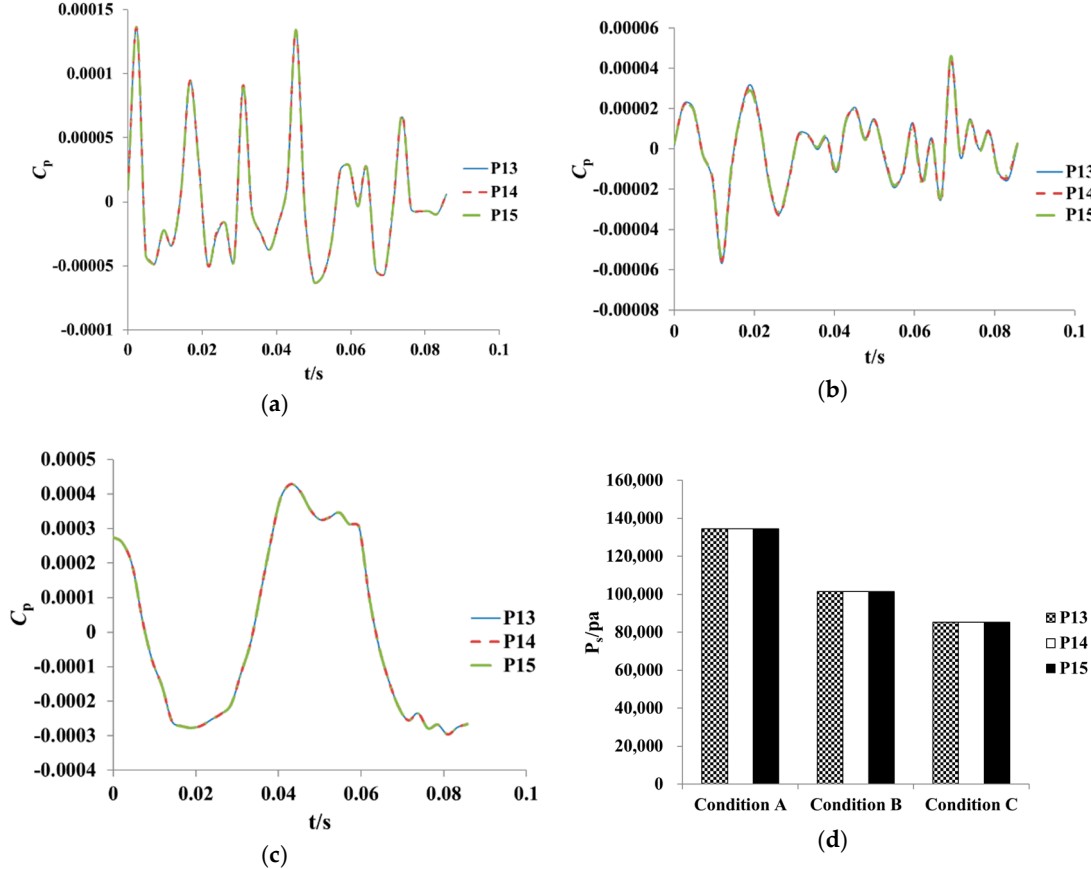

**Figure 21.** Pressure pulsation time domain diagram and pressure pulsation amplitude (P13–P15). (**a**) Condition A (best efficiency point). (**b**) Condition B (start point of hydraulic unstable zone). (**c**) Condition C (valley point of hydraulic unstable zone). (**d**) Pressure pulsation amplitude.

**Table 2.** Frequency domains of P13–P15.

| Condition | Parameters | P13 | | P14 | | P15 | |
|---|---|---|---|---|---|---|---|
| | | **MF** | **SF** | **MF** | **SF** | **MF** | **SF** |
| Condition A | $f$/Hz | 26.25 | 40.83 | 26.25 | 40.83 | 26.25 | 40.83 |
| | $T_f$ | 2.25 | 3.50 | 2.25 | 3.50 | 2.25 | 3.50 |
| | $P_s$/Pa | 584.39 | 192.31 | 584.41 | 192.31 | 584.43 | 192.32 |
| Condition B | $f$/Hz | 26.25 | 40.83 | 26.25 | 40.83 | 26.25 | 40.83 |
| | $T_f$ | 2.25 | 3.50 | 2.25 | 3.50 | 2.25 | 3.50 |
| | $P_s$/Pa | 441.86 | 145.12 | 441.86 | 145.13 | 441.87 | 145.13 |
| Condition C | $f$/Hz | 26.25 | 40.83 | 26.25 | 40.83 | 26.25 | 40.83 |
| | $T_f$ | 2.25 | 3.50 | 2.25 | 3.50 | 2.25 | 3.50 |
| | $P_s$/Pa | 380.26 | 120.18 | 380.26 | 120.18 | 380.27 | 120.18 |

Under the same condition, the pressure pulsation trend of monitoring probes P13–P15 is completely unanimous, but the amplitudes increase sequentially. This indicates that the farther away from the pump shaft, the smaller the pressure pulsation amplitude, the closer to the pump shaft, and the greater the pressure pulsation amplitude due to the stream disturbance. Under different conditions, the pressure pulsation trend on each monitoring probe is quite different, accompanied by poor periodicity, small flow rate, and pressure pulsation amplitude.

The main frequency and secondary frequency of each monitoring probe are 26.25 Hz and 40.83 Hz, corresponding to the 2.25 times and 3.5 times shaft frequency. For the same monitoring probe, great variation exists between the pressure pulsation amplitude of main frequency (MF) and secondary

frequency (SF). The pressure pulsation amplitude of MF is three times that of SF. Under different monitoring probes, the pressure pulsation amplitudes of MF and SF decrease as the flow rate reduces and small variation occurs between the pressure pulsation amplitude of MF and SF.

Figure 22 shows the pressure pulsation time domain diagram and comparison of pressure pulsation amplitude of the monitoring probes P10–P12 in a period under different conditions. Table 3 lists the main frequency, the secondary frequency, and the corresponding pressure amplitude of the monitoring probes P10–P12 under different conditions.

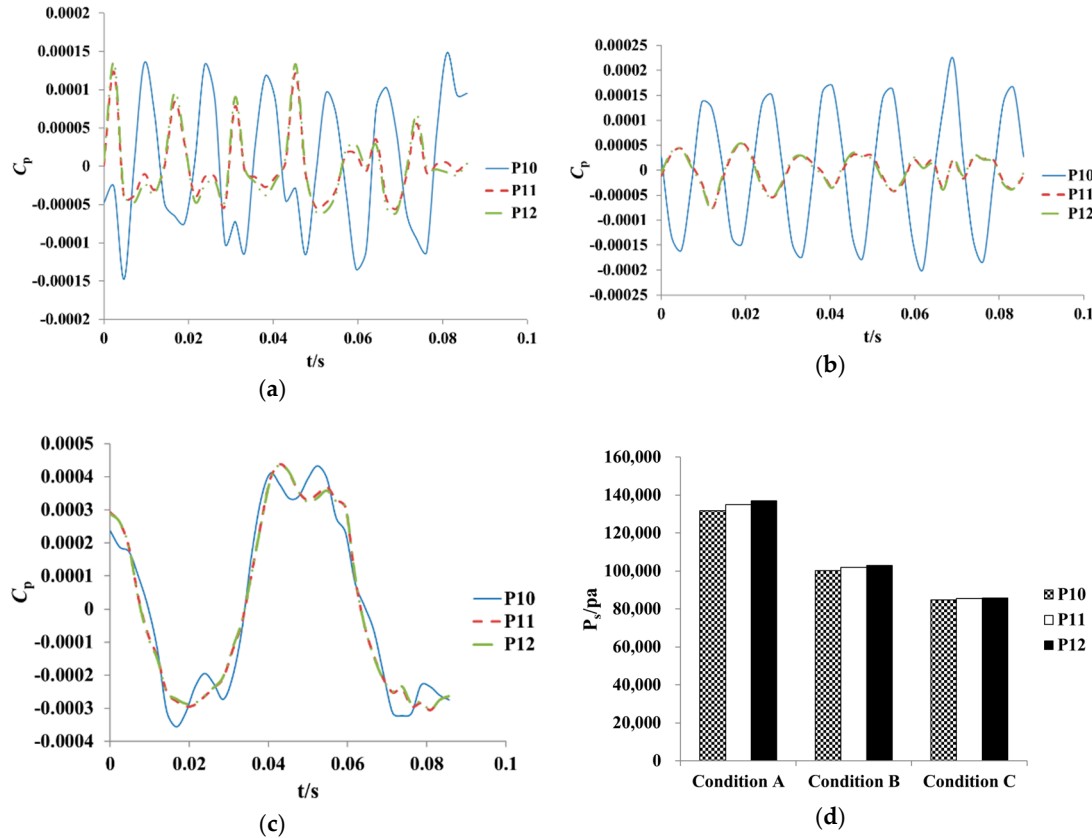

**Figure 22.** Pressure pulsation time domain diagram and pressure pulsation amplitude (P10–P12). (**a**) Condition A (best efficiency point). (**b**) Condition B (start point of hydraulic unstable zone). (**c**) Condition C (valley point of hydraulic unstable zone). (**d**) Pressure pulsation amplitude.

**Table 3.** Frequency domains of P10–P12.

| Condition | Parameters | P10 | | P11 | | P12 | |
|---|---|---|---|---|---|---|---|
| | | MF | SF | MF | SF | MF | SF |
| Condition A | $f$/Hz | 26.25 | 40.83 | 26.25 | 40.83 | 26.25 | 40.83 |
| | $T_f$ | 2.25 | 3.50 | 2.25 | 3.50 | 2.25 | 3.50 |
| | $P_s$/Pa | 572.23 | 188.36 | 586.94 | 193.14 | 595.22 | 195.86 |
| Condition B | $f$/Hz | 26.25 | 40.83 | 26.25 | 40.83 | 26.25 | 40.83 |
| | $T_f$ | 2.25 | 3.50 | 2.25 | 3.50 | 2.25 | 3.50 |
| | $P_s$/Pa | 435.41 | 142.97 | 443.21 | 145.57 | 447.65 | 147.03 |
| Condition C | $f$/Hz | 26.25 | 40.83 | 26.25 | 40.83 | 26.25 | 40.83 |
| | $T_f$ | 2.25 | 3.50 | 2.25 | 3.50 | 2.25 | 3.50 |
| | $P_s$/Pa | 377.66 | 119.41 | 380.65 | 120.35 | 382.49 | 120.94 |

The pressure pulsation amplitude increases from P10 to P12 under the same condition. The pressure pulsation amplitude at P12 is 1.04, 1.03, and 1.01 times of P10 under condition A, B, and C. The closer

to the shaft, the greater pressure pulsation amplitude, meaning the pressure pulsation amplitude is affected by the shaft. Under different conditions, the pressure pulsation amplitude and flow rate of each monitoring probe shows a positive relationship.

Under condition A, B, and C the MF and SF on the outlet of prolonged inlet section is completely the same with those in the prolonged inlet section. The rotating impeller has not yet been able to exert a dominant influence on the flow on the outlet of prolonged inlet section. Such a frequency shows no relationship with the blade frequency. The pressure pulsation amplitudes of MF and SF enlarges from P10 to P12 and the pressure pulsation amplitude of MF is also three times of SF.

The pressure pulsation time domain diagram and comparison of pressure pulsation amplitude of the monitoring probes P7–P9 in a period under different conditions is shown in Figure 23. The main frequency, the secondary frequency and the corresponding pressure amplitude of the monitoring probes P7–P9 under different conditions are listed in Table 4.

In general, the effect of the impeller on the inflow starts from the stream entering into the impeller, but actually the effect of the blade on the inflow begins when the water flow does not enter the impeller, mainly manifested by the pre-spin action on the water flow. Under the same condition, the pressure pulsation amplitude on the monitoring probes P7–P9 is gradually reduced, where P7 is the monitoring probe near the shroud and P9 is the monitoring probe near the hub. Thus, the pressure pulsation amplitude gradually increases from the hub to the shroud. The pressure pulsation amplitude at the monitoring probe P7 is 1.02 times, 1.05 times, and 1.01 times of P9 under condition A, B, and C. Hence, the pressure pulsation amplitude on each monitoring probe is positively correlated with the flow rate under different working conditions.

**Table 4.** Frequency domains of P7–P9.

| Condition | Parameters | P7 | | P8 | | P9 | |
|---|---|---|---|---|---|---|---|
| | | MF | SF | MF | SF | MF | SF |
| | $f$/Hz | 70 | 140 | 70 | 26.25 | 26.25 | 70 |
| Condition A | $T_f$ | 6 | 12 | 6 | 2.25 | 2.25 | 6 |
| | $P_s$/Pa | 4600.42 | 1843.95 | 2115.09 | 587.59 | 593.25 | 444.94 |
| | $f$/Hz | 70 | 140 | 70 | 140 | 70 | 26.25 |
| Condition B | $T_f$ | 6 | 12 | 6 | 12 | 6 | 2.25 |
| | $P_s$/Pa | 2706.53 | 1260.64 | 1358.23 | 451.00 | 581.55 | 414.95 |
| | $f$/Hz | 70 | 140 | 70 | 26.25 | 70 | 26.25 |
| Condition C | $T_f$ | 6 | 12 | 6 | 2.25 | 6 | 2.25 |
| | $P_s$/Pa | 1100.13 | 370.35 | 986.14 | 349.29 | 385.55 | 357.89 |

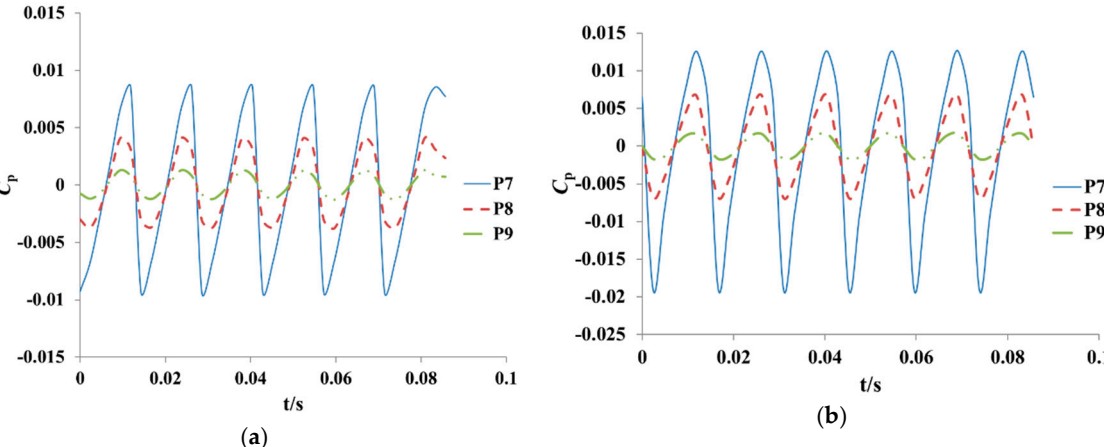

(a)                    (b)

**Figure 23.** *Cont.*

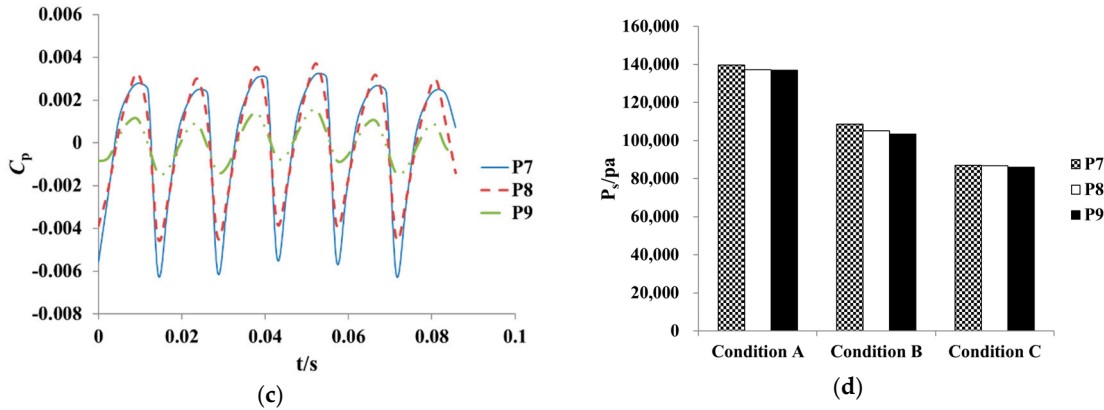

**Figure 23.** Pressure pulsation time domain diagram and pressure pulsation amplitude (P7–P9). (**a**) Condition A (best efficiency point). (**b**) Condition B (start point of hydraulic unstable zone). (**c**) Condition C (valley point of hydraulic unstable zone). (**d**) Pressure pulsation amplitude.

The frequency domain of monitoring probes set at the inlet of the impeller is observed and analyzed. The MF and SF on the impeller inlet monitoring probe are not certain, wherein the MF is 70 Hz or 26.25 Hz, and the SF is 70 Hz, 140 Hz, or 26.25 Hz. Under condition A, the MF of P7 and P8 is 70 Hz, which is the blade frequency, but the MF of P9 is 2.25 times of the shaft frequency. Under condition B, and C, the MF on each monitoring probe is also the blade frequency, but the SF is 26.25 Hz or 140 Hz. Under all conditions, the pressure pulsation amplitude of MF for P7 and P8 is quite different from the pressure pulsation amplitude of SF. However, the pressure pulsation amplitude of MF of P9 is basically consistent with SF. The monitoring probe P7 is pre-installed near the shroud, and the MF and SF are 70 Hz and 140 Hz, respectively, which is 1 and 2 times the blade frequency. Thus, the blade frequency plays a dominant role in the pressure pulsation near the shroud under condition A, B, and C. The location of monitoring probe P8 is between the shroud and the hub, and the MF is 70 Hz, the SF is 26.25 Hz under condition A and C, but is 140 Hz under condition B. Therefore, the blade frequency is still the dominant factor, but the shaft frequency also begins to play a certain role in it. The monitoring probe P9 is pre-set near the hub. Thus, 26.25 Hz and 70 Hz occur alternately in the MF and the SF of P9, and the pressure pulsation amplitudes of the MF is matched roughly with the SF. This indicates that the pressure pulsation near the hub is affected by both the impeller and the pump shaft.

Figure 24 shows the pressure pulsation time domain diagram and comparison of pressure pulsation amplitude of the monitoring probes P4–P6 in a period under different conditions. Table 5 lists the main frequency, the secondary frequency, and the corresponding pressure amplitude of the monitoring probes P4–P6 under different conditions.

Under the same condition, the pressure pulsation amplitude of P4–P6 decreases successively. Thus, the pressure pulsation amplitude near the shroud of the impeller inlet is higher than near the hub of the impeller outlet. Under condition A, B, and C, the pressure pulsation amplitude of P4 is 1.03 times, 1.04 times, and 1.07 times of P6. Under different conditions, the pressure pulsation amplitude decreases with the reducing flow rate, which indicates that the pressure pulsation amplitude has a positive correlation with the flow rate.

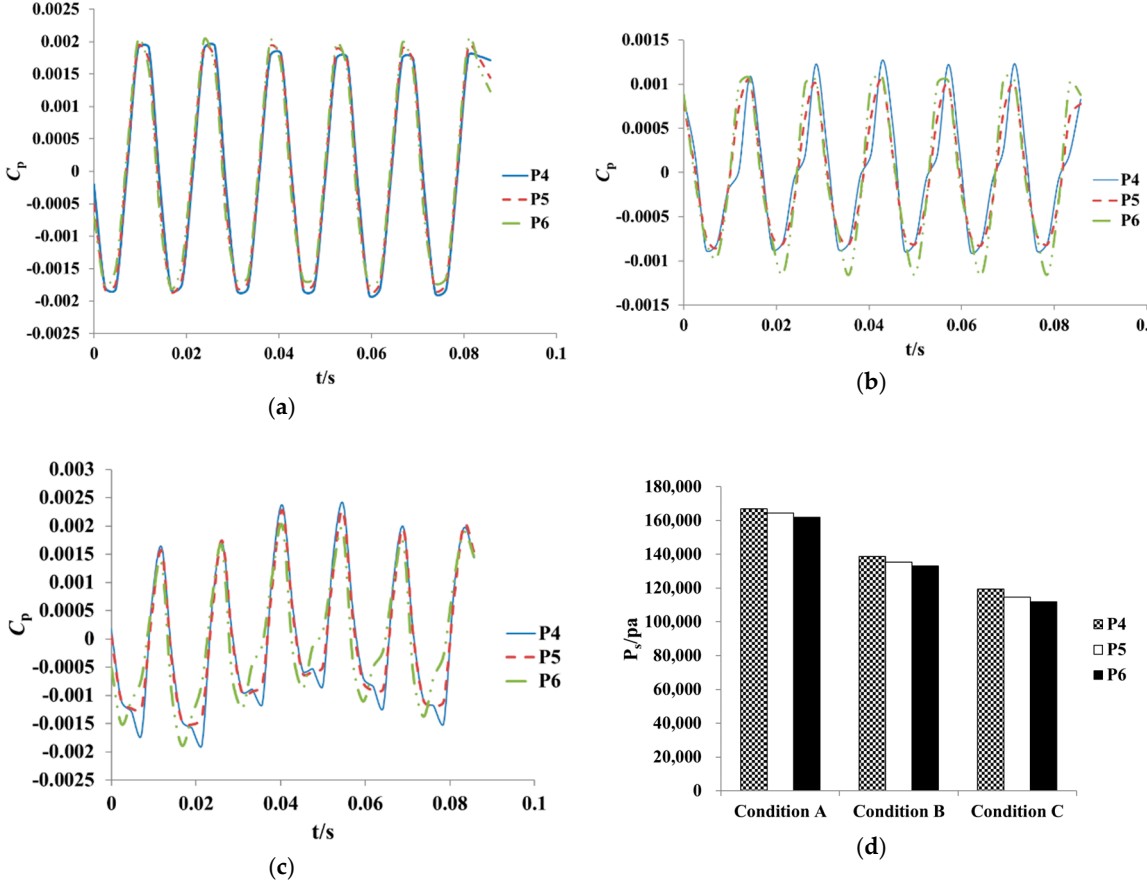

**Figure 24.** Pressure pulsation time domain diagram and pressure pulsation amplitude (P4–P6). (**a**) Condition A (best efficiency point). (**b**) Condition B (start point of hydraulic unstable zone). (**c**) Condition C (valley point of hydraulic unstable zone). (**d**) Pressure pulsation amplitude.

**Table 5.** Frequency domains of P4–P6.

| Condition | Parameters | P4 | | P5 | | P6 | |
|---|---|---|---|---|---|---|---|
| | | MF | SF | MF | SF | MF | SF |
| Condition A | $f$/Hz | 26.25 | 70 | 26.25 | 70 | 26.25 | 70 |
| | $T_f$ | 2.25 | 6 | 2.25 | 6 | 2.25 | 6 |
| | $P_s$/Pa | 720.35 | 680.70 | 710.37 | 680.93 | 699.73 | 664.23 |
| Condition B | $f$/Hz | 26.25 | 70 | 26.25 | 70 | 26.25 | 70 |
| | $T_f$ | 2.25 | 6 | 2.25 | 6 | 2.25 | 6 |
| | $P_s$/Pa | 601.99 | 304.07 | 587.76 | 301.68 | 577.25 | 372.17 |
| Condition C | $f$/Hz | 70 | 26.25 | 26.25 | 70 | 26.25 | 70 |
| | $T_f$ | 6 | 2.25 | 2.25 | 6 | 2.25 | 6 |
| | $P_s$/Pa | 524.16 | 520.84 | 499.13 | 486.14 | 482.78 | 425.16 |

Under condition A, the MF and SF of each monitoring probe are 26.25 Hz and 70 Hz. The pressure pulsation amplitude corresponding to MF is slightly higher than SF. Under condition B, the MF and the SF of each monitoring probe are also 26.25 Hz and 70 Hz. For monitoring probes P4–P6, the pressure pulsation amplitude of MF is 1.98 times, 1.95 times, and 1.55 times of SF. Under condition C, the MF and SF of the monitoring probe P4 are 70 Hz and 26.25 Hz. The MF and SF of the monitoring probes P5 and P6 are 26.25 Hz and 70 Hz, and the pressure pulsation amplitudes of MF and SF are consistent.

Figure 25 shows the pressure pulsation time domain diagram and comparison of pressure pulsation amplitude of the monitoring probes P1–P3 in a period under different conditions. Table 6

lists the main frequency, the secondary frequency, and the corresponding pressure amplitude of the monitoring probes P1–P3 under different conditions.

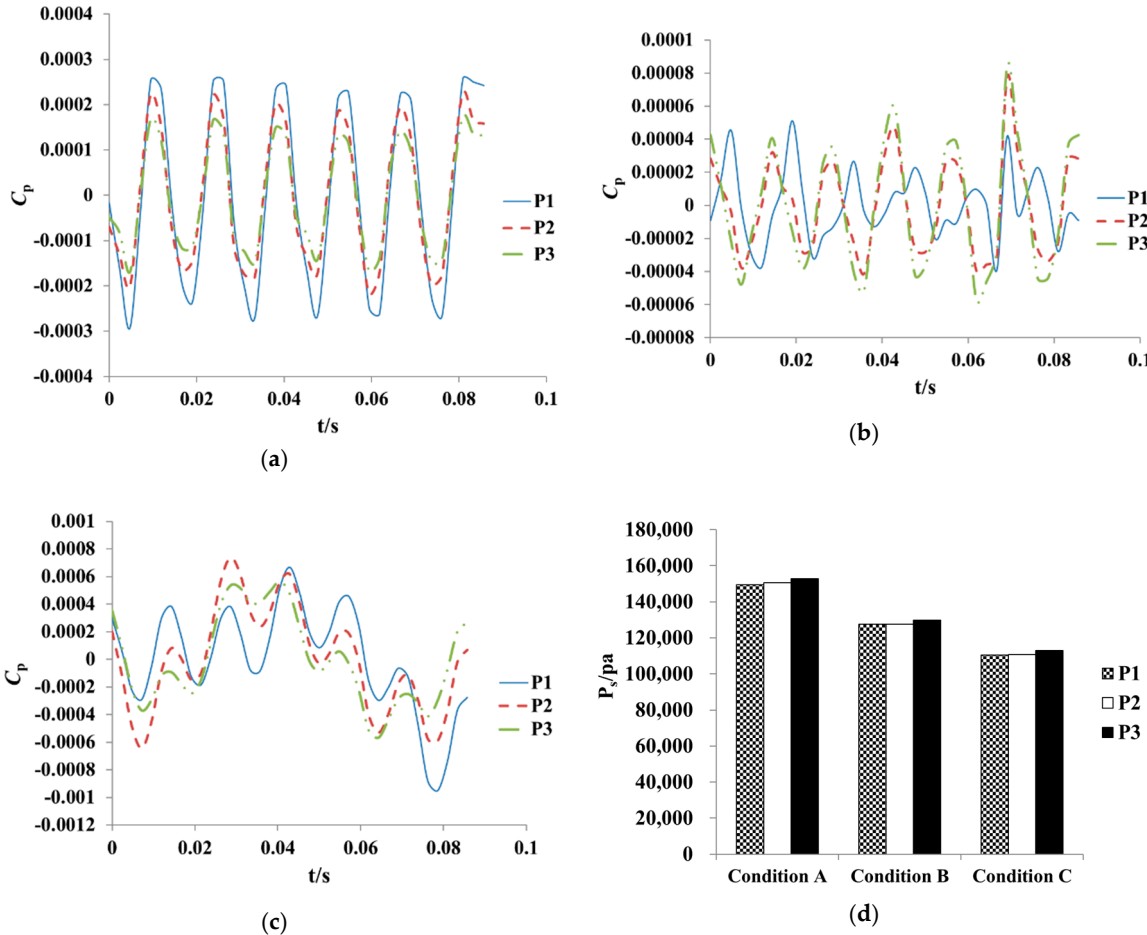

**Figure 25.** Pressure pulsation time domain diagram and pressure pulsation amplitude (P1–P3). (**a**) Condition A (best efficiency point). (**b**) Condition B (valley point of hydraulic unstable zone). (**c**) Condition C (valley point of hydraulic unstable zone). (**d**) Pressure pulsation amplitude.

**Table 6.** Frequency domains of P1–P3.

| Condition | Parameters | P1 | | P2 | | P3 | |
|---|---|---|---|---|---|---|---|
| | | MF | SF | MF | SF | MF | SF |
| Condition A | $f$/Hz | 26.25 | 40.83 | 26.25 | 40.83 | 26.25 | 40.83 |
| | $T_f$ | 2.25 | 3.50 | 2.25 | 3.50 | 2.25 | 3.50 |
| | $P_s$/Pa | 649.35 | 213.56 | 654.13 | 215.19 | 664.25 | 218.49 |
| Condition B | $f$/Hz | 26.25 | 40.83 | 26.25 | 40.83 | 26.25 | 40.83 |
| | $T_f$ | 2.25 | 3.50 | 2.25 | 3.50 | 2.25 | 3.50 |
| | $P_s$/Pa | 554.60 | 182.05 | 555.11 | 182.15 | 565.04 | 185.43 |
| Condition C | $f$/Hz | 29.62 | 41.46 | 29.17 | 40.83 | 29.17 | 40.83 |
| | $T_f$ | 2.53 | 3.55 | 2.50 | 3.50 | 2.50 | 3.50 |
| | $P_s$/Pa | 488.05 | 157.02 | 486.74 | 156.61 | 497.70 | 156.84 |

Under the same condition, the pressure pulsation amplitudes of the monitoring probes P1–P3 increase sequentially. The pressure pulsation amplitude near the hub of the GVs outlet is higher than it near the shroud of the GVs outlet. Under condition A, B, and C, the pressure pulse

amplitude of P1 is 1.02 times of P3. The pressure pulsation amplitude on the same monitoring probe under different conditions is positively correlated with the flow rate.

Under condition A, the MF and the SF of P1–P3 are 26.25 Hz and 40.83 Hz, which are 2.25 times and 3.50 times of the shaft frequency. The pressure pulsation amplitude of the MF is about 3 times of SF. Under condition B, the MF and the SF of P1–P3 are also 26.25 Hz and 40.83 Hz. The pressure pulsation amplitude of the MF is also about 3 times of SF. Under condition C, the MF and SF of P1 are 29.62 Hz and 41.46 Hz. The MF and SF of P2 are 29.17 Hz and 40.83 Hz. The MF and SF of P3 are 29.17 Hz and 40.83 Hz. The MF and SF of P1–P3 are no longer consistent and lacks of regularity. On one hand, the rotation of impeller and shaft affects the flow downstream a little for the outlet of GV is far from the impeller. On the other hand, the flow pattern in the propulsion pump is not stable and the MF and the SF on each monitoring probe are fluctuating in the low frequency range.

## 5. Conclusions

(1) The mixed-flow waterjet propulsion device is tested by establishing the double circulation test loop of waterjet propulsion system. The test results are consistent with CFD results both in the trend and values. The CFD method is reliable.

(2) Conditions A, B, and C are marked as characteristic conditions by analyzing the hydraulic performance of the propulsion pump, which are the BEP (best efficiency point), start point of hydraulic unstable zone, and the valley point of hydraulic unstable zone. Thus, unsteady calculation is promoted and the unsteady flow process of the propulsion pump at different times of the same period is discussed. The surface vortex on the blade under condition C is unstable, and the vortex core and shape pattern vary on a small scale as time. Three turbo surfaces are sliced to study the flow features on each spanwise under different conditions. The steady flow characteristic of each turbo surface is obtained under condition A and B; however, the flow characteristic of each turbo surface varies as time under condition C, due to the unstable velocity and pressure field.

**Author Contributions:** Data curation, L.C. and W.J.; Formal analysis, C.W.; Methodology, C.L.; Writing—original draft, H.L.; Writing—review & editing, D.Z.

**Funding:** This research was funded by [Jiangsu Province Science Foundation for Youths] grant number [BK20170507], [Natural Science Foundation of the Jiangsu Higher Education Institutions] grant number [17KJD580003], [Jiangsu Planned Projects for Postdoctoral Research Funds] grant number [1701189B], [Open Research Subject of Key Laboratory of Fluid and Power Machinery (Xihua University), Ministry of Education] grant number [szjj2019-018], [Science and Technology Innovation and Cultivation Fund of Yangzhou University] grant number [2017CXJ047], [National Natural Science Foundation of China] grant number [51779214], [Priority Academic Program Development of Jiangsu Higher Education Institutions (PAPD)], [Jiangsu Province 333 high level talents training project] grant number [(2018) III-1827], [Peak plan six talents in Jiangsu province], and [Key project of water conservancy in Jiangsu province] [2018042].

**Conflicts of Interest:** The authors declare no conflict of interest.

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
