# Peer review of "Unsteady Flow Process in Mixed Waterjet Propulsion Pumps with Nozzle Based on Computational Fluid Dynamics"

_processes, doi:10.3390/pr7120910_

Round 1

Reviewer 1 Report

In this paper, the authors analyzed the unsteady flow process in mixed waterjet propulsion pump with nozzle at the unsteady condition. Their CFD simulation results demonstrate the surface vortex of the blade is unstable at the valley conditions of the hydraulic unsteady zone, and the vortex core and morphological characteristics of the vortex will change in a small range with time. The results may have some implications in the design of mechanical pumps. I recommend it to be published on Processes after addressing the following issues.

1) It is still not clear for me what the motivation and what the significance of this work is? So I recommend the authors to add one sentence to demonstrate the motivation/background and one sentence about the possible application in Abstract.

2) In the Introduction part, the authors mention that “The water jet propulsion device has the advantages of flexible operation, excellent maneuverability, high speed, outstanding anti-cavitation performance and high efficiency”. Suitable citations or sources should be given here.

3) Recently, bioinspired valveless pumps also becomes a very hot topics. I highly recommend the authors to briefly describe it in the Introduction to improve the broad interests of future readers. Two nice papers are recommended for the authors to cite, such as Science312(5774), pp.751-753, and Proceedings of the National Academy of Sciences116(5), pp.1543-1548.

 4) Another big issues in the main text is the figure captions. In figure 3, authors should add sub figure indexes and then describe both of them separately and clearly. The same problems are for the figure 4.  In figure 8-10, the authors should briefly describe the Conditions A, B and C, thus the readers do not need to go back to the main text to figure out what they are? Same issued are found in other figures as well.

Author Response

1) It is still not clear for me what the motivation and what the significance of this work is? So I recommend the authors to add one sentence to demonstrate the motivation/background and one sentence about the possible application in Abstract.

Response: The authors wish to thank the Reviewer’s recommendation. Two sentences are added in the Abstract.

To demonstrate the motivation/background, one sentence is added as below (in Palatino Linotype red font of the latest upload manuscript): “The unsteady flow process is related to the comprehensive performance and phenomenon such as rotating stall and cavitation in the waterjet pump.”

Another sentence about the possible application is added as below (in Palatino Linotype red font of the latest upload manuscript): “The outcome in this paper is helpful to comprehend the unsteady flow mechanism in the pump of the waterjet propulsion device, then improve and benefit the design and comprehensive performance of it.”

2) In the Introduction part, the authors mention that “The water jet propulsion device has the advantages of flexible operation, excellent maneuverability, high speed, outstanding anti-cavitation performance and high efficiency”. Suitable citations or sources should be given here.

Response: The authors wish to thank the Reviewer’s suggestions. One suitable source “[1] John Allison. Marine Waterjet Propulsion [J]. SNAME Transactions. 1993(101): 275-335.” is given in References (in Palatino Linotype red font of the latest upload manuscript).

3) Recently, bioinspired valveless pumps also becomes a very hot topics. I highly recommend the authors to briefly describe it in the Introduction to improve the broad interests of future readers. Two nice papers are recommended for the authors to cite, such as Science, 312(5774), pp.751-753, and Proceedings of the National Academy of Sciences, 116(5), pp.1543-1548.

Response: The authors wish to thank the Reviewer’s comments. Descriptions about bioinspired valveless pump are added in the Introduction to improve the broad interests of future readers, as below (in Palatino Linotype red font of the latest upload manuscript): “Moreover, the pump is essential for all animals. The heart tube of the embryonic vertebrate has been described as a peristaltic pump before the development of discernable chambers and valves at these early stages [6]. Then the bioinspired valveless pump is designed due to the pumping process of the heart tube and applied in the fields such as microfluidics, drug delivery, biomedical devices, cardiovascular pumping system, becoming a very hot topic nowadays [7].” And the two corresponding references have been cited (in Palatino Linotype red font of the latest upload manuscript).

[6]   Arian S. Forouhar, Michael Liebling, Anna Hickerson, Abbas Nasiraei-Moghaddam, Huai-Jen Tsai, Jay R. Hove, Scott E. Fraser, Mary E. Dickinson, Morteza Gharib. The Embryonic Vertebrate Heart Tube Is a Dynamic Suction Pump [J]. Science. 2006, 312(5774): 751-753.

[7]   Zhengwei Li, Yongbeom Seo, Onur Aydin, Mohamed Elhebeary, Roger D. Kamm, Hyunjoon Kong, and M. Taher A. Saif. Biohybrid valveless pump-bot powered by engineered skeletal muscle [J]. PROCEEDINGS OF THE NATIONAL ACADEMY OF SCIENCES OF THE UNITED STATES OF AMERICA. 2019, 116(5): 1543-1548.

 4) Another big issues in the main text is the figure captions. In figure 3, authors should add sub figure indexes and then describe both of them separately and clearly. The same problems are for the figure 4.  In figure 8-10, the authors should briefly describe the Conditions A, B and C, thus the readers do not need to go back to the main text to figure out what they are? Same issued are found in other figures as well.

Response: The authors wish to thank the Reviewer’s comments. Sub figure indexes and the description of them are added in figure 3 and 4. Descriptions of condition A, B and C are added as “best efficiency point”, “start point of hydraulic unstable zone” and “valley point of hydraulic unstable zone” in figure 8~10 and figure 12~25 (in Palatino Linotype red font of the latest upload manuscript).

Reviewer 2 Report

This paper presents a numerical simulation result regarding the hydraulic performance and unsteady flow process of a propulsion pump with nozzle. The authors succeed in validating their computational capability through a comparison of the CFD result with model test result.

However, the lack of a systematic presentation for the results obtained from CFD which typically outputs an abundant information, is found in the manuscript. Followings are comments minimally made from that point of view:

(1) Numerical methodolgy which was employed in the simulation should be appropriately described for the fruitful information service to the readers, even though the CFD method was proven to be reliable. The manuscript includes only a grid system and its validation results in the relevant section.  

(2) The description for the figures and tables should be made as concisely as possible. Repeating the similar statements for the figures and tables should be minimized. Line#141~146 can be tabulized, rather than those repetitive descriptions, for more clear illustration along with the relevant figures. Likewise, the discussions on the results should be rearranged or reformatted in order to get out of the repetitive and tedious statements for the very similar figures and phenomena.

(3) Some errors are found in English translation. For example, in Line#116, 'scheme' is more acceptable word than 'style' in CFD community. Attention should be also paid to spacing words between numeral and unit.

-The end-

Author Response

(1) Numerical methodolgy which was employed in the simulation should be appropriately described for the fruitful information service to the readers, even though the CFD method was proven to be reliable. The manuscript includes only a grid system and its validation results in the relevant section.

Response: The authors wish to thank the Reviewer’s comments. Numerical methodolgy about the governing equation and calculation method are replenished in section 2.3 as below: “Due to no heat transfer exists in the flow process of waterjet propulsion device, Navier Stokes equations are utilized as the governing equations to describe the flow. And the finite volume method is also adopted to calculate the flow in the computational domains.” (in Palatino Linotype red font of the latest upload manuscript)

(2) The description for the figures and tables should be made as concisely as possible. Repeating the similar statements for the figures and tables should be minimized. Line#141~146 can be tabulized, rather than those repetitive descriptions, for more clear illustration along with the relevant figures. Likewise, the discussions on the results should be rearranged or reformatted in order to get out of the repetitive and tedious statements for the very similar figures and phenomena.

Response: The authors wish to thank the Reviewer’s comments. Line#141~146 is tabulized as Tab.1 (in Palatino Linotype red font of the latest upload manuscript) listing the locations of each probe, instead of the previous word description.

Tab.1 Locations of each probe

Number

Location

P1~P3

Outlet of GV

P4~P6

Outlet of impeller

P7~P9

Inlet of impeller

P10~P12

Outlet of the prolonged inlet section

P13~P15

In the prolonged inlet section

According to the suggestion, the discussions on the results are rearranged or reformatted. In the previous version, the results of different turbo surfaces at different conditions are given as condition A, B and C. The latest version (in Palatino Linotype red font of the latest upload manuscript) adopts turbo surface TS1, TS2 and TS3 by deleting or rewriting some similar descriptions and avoiding repetitive and tedious statements, as below:

“Fig.12~14 are the streamlines on turbo surfaces TS1 at the condition A, B and C. In which, the velocity in the impeller is the relative velocity and the velocity in the GV is the absolute velocity. The streamlines on the turbo surfaces of the impeller and GV varies slightly with time, therefore, the flow is steady. At condition A, velocity on the suction side of impeller is high, the inlet attack angle is basically the same as the angle of the leading edge of the airfoil, leading to the excellent inflow condition and smooth stream in the blade-to-blade passage. Small scale spanning vortex occurs at the tailing edge of the suction. Due to the adjustment of the GV and the location of vortex away from the impeller, the geometric shape and magnitude of the vortex shows no evident relationship with the time. At condition B, slight deviation exists between the inlet attack angle of the impeller and the GV and the airfoil angle of the blade. The low velocity region of the pressure surface of the leading edge of the impeller blade is enlarged. Large-scale spanning vortex, extending from the inlet to the outlet in the axial direction and occupying about 1/3 of the groove in the spanning direction occurs in each groove of GV. Due to the spanning vortex, the streamlines of other parts in the groove are severely skewed and then gathers near the outlet of the GV. At condition C, smooth streamline in the groove is mildly affected. Distinct spanning vortex in the groove still exists and covers half of the groove on the spanning direction. The status and range of spanning vortex are basically maintained. But the vertex core migrates in a small scale and the vortex status modifies, meaning the flow characteristic of GV is unstable.”

“Fig.15~17 are the streamlines on turbo surfaces TS2 at the condition A, B and C. At condition A, streamlines in the blade-to-blade passage of impeller and GV is smooth, no vortex happens. At condition B, slight deviation also happens, but the streamlines in the groove of the impeller is smooth. As the arrows in Fig.16, the streamlines near the tailing edge on the suction side of the impeller deviates slightly. Spanning vortex in the groove of GV disappears, but the shedding vortex is observed on the tailing edge of the outlet in the impeller, resulting in part of streamlines severely skewed which has a little impact on the mainstream. At condition C, the inlet attack angle of the impeller is substantially the same as the airfoil angle of the blade. Distinct spanning vortex is observed near the tailing of different blades. The variation between the inlet attack angle of GV and the airfoil angle of the blade is apparent. Three vortexes marked as SV1, SV2 and SV3 happens on the spanning surface of GV. SV1 and SV2, located at the head edge and tailing edge of suction side, are in the channel of groove and rotates in the clockwise. The range of SV1 is much larger than SV2. The shedding vortex SV3 located at the tailing of GV twirls on the opposite direction of the spanning vortex. As the time passes, the shape and range of SV1, SV2 and SV3 varies.”

“Fig.18~20 are the streamlines on turbo surfaces TS2 at the condition A, B and C. At condition A, the inlet attack angle is not the same as the angle of the leading edge of the airfoil. In one blade-to-blade passage, part of stream flows from the suction side to pressure side, then out of the blade-to-blade passage along the pressure side, and the streamlines converge at the tailing edge of airfoil. Flow in the blade-to-blade passage of GV is similar to the impeller. At condition B, the steam at the entrance of the impeller flows into the neighboring groove on the opposite rotating direction instead into the GV. Part of the steam at the head edge on the suction side, dragged by the high speed steam at the entrance of the groove, flows to the head edge on the pressure side of the neighboring blade, then outside of the groove and into the neighboring groove. Low velocity zone exists in the groove of the impeller and the streamline is disordered. At condition C, the distinction between the inlet attack angle of the impeller and the airfoil angle of the blade is huge and obvious spanning vortex appears at the head edge of the pressure side and the tailing edge of the suction side. Both spanning vortexes nearly cover the whole channel of the groove in the spanning direction. As the time passes, the shape of spanning vortex will change, but the vortex core will not migrate and the location maintains.”

(3) Some errors are found in English translation. For example, in Line#116, 'scheme' is more acceptable word than 'style' in CFD community. Attention should be also paid to spacing words between numeral and unit.

Response: As suggested, 'style' is replaced with 'scheme' (in Palatino Linotype red font of the latest upload manuscript). Spacing words between numeral and unit are added in all corresponding locations of the manuscript.

Round 2

Reviewer 2 Report

This paper presents a numerical simulation result regarding the hydraulic performance and unsteady flow process of a propulsion pump with nozzle. The authors succeed in validating their computational capability through a comparison of the CFD result with model test result.

Corrective action was appropriately taken to the points raised in the previous review.

Moderate English changes are still required.

-The End-

Author Response

This paper presents a numerical simulation result regarding the hydraulic performance and unsteady flow process of a propulsion pump with nozzle. The authors succeed in validating their computational capability through a comparison of the CFD result with model test result.

Corrective action was appropriately taken to the points raised in the previous review.

Moderate English changes are still required.

Response:

The authors wish to thank the Reviewer’s comments. English expressions are improved as below (in Palatino Linotype red font of the latest upload manuscript).

In Abstract:

Line 11~12

“The unsteady flow process of waterjet pump is related to the comprehensive performance and phenomenon such as rotating stall and cavitation.”

Line 17~20

“At the valley point of hydraulic unstable zone, the flow and pressure fields are unstable which causes the flow on each turbo surface changes with time. The hydraulic performance parameters are measured by establishing the double cycle test loop of waterjet propulsion device and compared with the numerical simulated data.”

In Introduction

Line 31~32

“The propulsion pump is well protected for being arranged inside.”

Line 36~38

“Considering the thrust and layout requirements, the guide vane mixed flow pump and axial flow pump are generally adopted in the current waterjet propulsion vessels.”

Line 72~74

“Recently, the main works focused on the property of the waterjet propulsion device, but the research on the unsteady flow process between the nozzle and the propulsion pump has not been seen yet.”

In Numerical simulation method

Line 80~81

“A computational domain geometric model including the nozzle is established to obtain stable flow field data of the mixed flow waterjet propulsion device.”

Line 92~93

“However, if the number of meshes is too large, it will occupy numerous computing resources.”

Line 98~99

“The head H and efficiency η is calculated by Eqs. (1) and (2) when the flow rate is QBEP, 1.13QBEP and 1.33QBEP.”

Line 107

“The relative head and relative efficiency are darwn in Fig.3.”

Line 111~112

“this trend exceeds to clearer as the flow rate increases”

Line 123

“As heat transfer does not exist in the flow process of waterjet propulsion device”

Line 128

“In order to guarantee the value transfer”

Line 147~149

“In terms of the regular pressure pulsation, the main induced factors are the impeller rotation, the pump shaft rotation and rotor-stator interaction, which are respectively related to the axial frequency and the blade frequency.”

Line 152~154

“The observed pressure pulsation characteristics are distinct for different pumps, and for the same pump, the observed pressure pulsation characteristics are diverse under various operating conditions and monitoring positions.”

Line 161~162

“Data of monitoring parameters are stored when the impeller rotates per degree”

Line 165~166

“The data of the 8th period on each monitoring probes are plotted into the time domain chart of pressure pulsation.”

Line 168~170

“The time domain chart is utilized in the time domain analysis method. In the chart, the abscissa is the time-related parameters such as the time and period and the ordinate is the pressure.”

Line 174~175

“The frequency domain analysis illustrates the main pulsating component of the pressure pulsation and the primary factor affecting the pressure pulsation directly.”

Line 178

“For the impeller rotates at 700 rev/min”

In Test arrangement and verification

Line 189~193

“The main cycle, which is applied to provide the navigation speed for the waterjet propulsion pump, consists of centrifugal auxiliary pump, electromagnetic flow meter, butterfly valve, expansion joint, rectifying device and piping system. The secondary cycle, which is used to test the hydraulic performance, includes test zone (mixed pump), electromagnetic flow meter, butterfly valve and piping system.”

Line 201~202

“The CFD result shows a consistent trend with the test result. Therefore, the numerical method is reliable and suitable for the following research work.”

In Results

Line 221~222

“Under the condition B, the stream at the leading edge flows to the trailing edge and the tip of the blade.”

Line 244~245

“Under condition A, the streamlines are steady and vary slightly with time on the turbo surfaces of the impeller and GV. The velocity is rapid on the suction side of impeller.”

Line 250~251

“Under condition B, slight deviation exists between the inlet attack angle and the airfoil angle of the blade in the impeller and the GV.”

Line 252~255

“Large-scale spanning vortex occurs in each groove of GV, which extends from the inlet to the outlet in the axial direction and occupies about 1/3 of the groove in the spanning direction. The streamlines of other parts in the groove are severely skewed and then gathers near the outlet of the GV because of the spanning vortex.”

Line 269~273

“Under condition A, streamlines are smooth and no vortex happens in the blade-to-blade passage of impeller and GV. Under condition B, slight deviation also happens, but the streamlines are smooth in the groove of the impeller. As the arrows in Fig.16, the streamlines deviate slightly near the tailing edge on the suction side of the impeller. Spanning vortex disappears in the groove of GV”

Line 274~276

“Part of streamlines are severely skewed and have a little impact on the mainstream. Under condition C, the inlet attack angle is consistent with the airfoil angle of the blade in the impeller.”

Line 280~281

“SV3 is the shedding vortex and located at the tailing of GV twirls on the opposite direction of the spanning vortex.”

Line 293~294

“Under condition A, the inlet attack angle is inconsistent with the angle of the leading edge of the airfoil.”

Line 297~301

“The flow pattern in the blade-to-blade passage of GV is similar to the impeller. Under condition B, the steam at the entrance of the impeller flows into the neighboring groove on the opposite rotating direction instead of the GV. Part of the steam at the head edge on the suction side flows to the head edge on the pressure side of the neighboring blade and then out of the groove and into the neighboring groove for being dragged by the high speed steam at the entrance of the groove.”

Line 302~304

“Under condition C, the distinction is huge between the inlet attack angle of the impeller and the airfoil angle of the blade and obvious spanning vortex appears at the head edge of the pressure side and the tailing edge of the suction side.”

Line 306~307

“The shape of spanning vortex will change with time, but the vortex core will not migrate and the location maintains.”

Line 327~328

“Under the same condition, the pressure pulsation trend of monitoring probes P13~P15 is completely unanimous, but the amplitudes increase sequentially.”

Line 336~338

“The pressure pulsation amplitude of MF is three times that of SF. Under different monitoring probes, the pressure pulsation amplitudes of MF and SF decrease as the flow rate reduces and small variation happens between the pressure pulsation amplitude of MF and SF.”

Line 344~345

“The pressure pulsation amplitude at P12 is 1.04, 1.03 and 1.01 times of P10 under condition A, B and C.”

Line 364~366

“In general, the effect of the impeller on the inflow starts from the stream entering into the impeller, but actually the effect of the blade on the inflow begins when the water flow does not enter the impeller, mainly manifested by the pre-spin action on the water flow.”

Line 369~371

“The pressure pulsation amplitude at the monitoring probe P7 is 1.02 times, 1.05 times and 1.01 times of P9 under condition A, B and C.”

Line 384~385

“However, the pressure pulsation amplitude of MF of P9 is basically consistent with SF.”

Line 391

“The monitoring probe P9 is pre-set near the hub.”

Line 402~404

“Under condition A, B and C, the pressure pulsation amplitude of P4 is 1.03 times, 1.04 times, and 1.07 times of P6.”

Line 411~412

“The pressure pulsation amplitude corresponding to MF is slightly higher than SF.”

Line 413~414

“For monitoring probes P4~P6, the pressure pulsation amplitude of MF is 1.98 times, 1.95 times and 1.55 times of SF.”

Line 424~426

“The pressure pulsation amplitude near the hub of GV’s outlet is higher than it near the shroud of GV’s outlet. Under condition A, B and C, the pressure pulse amplitude of P1 is 1.02 times of P3.”

Line 433~435

“Under condition A, the MF and the SF of P1~P3 are 26.25Hz and 40.83Hz, which are 2.25 times and 3.50 times of the shaft frequency.”

“Under condition B, the MF and the SF of P1~P3 are also 26.25 Hz and 40.83 Hz.”

Line 438~440

“The MF and SF of P1~P3 are no longer consistent and lacks of regularity. On one hand, the rotation of impeller and shaft affects the flow downstream a little for the outlet of GV is far from the impeller.”

In Conclusions

Line 447~449

“Condition A, B and C is marked as characteristic conditions by analyzing the hydraulic performance of the propulsion pump, which are the BEP point, start point of hydraulic unstable zone and the valley point of hydraulic unstable zone.”
